# PRIVACY-PRESERVING MACHINE LEARNING FOR HEALTHCARE: OPEN CHALLENGES AND FUTURE PERSPECTIVES

**Alejandro Guerra-Manzanares**,* **L. Julian Lechuga Lopez**\*,
**Michail Maniatakos and Farah E. Shamout**
Department of Computer Engineering, New York University Abu Dhabi
{ag9454, ljl5178, mm6446, fs999}@nyu.edu

## ABSTRACT

Machine Learning (ML) has recently shown tremendous success in modeling various healthcare prediction tasks, ranging from disease diagnosis and prognosis to patient treatment. Due to the sensitive nature of medical data, privacy must be considered along the entire ML pipeline, from model training to inference. In this paper, we conduct a review of recent literature concerning Privacy-Preserving Machine Learning (PPML) for healthcare. We primarily focus on privacy-preserving training and inference-as-a-service, and perform a comprehensive review of existing trends, identify challenges, and discuss opportunities for future research directions. The aim of this review is to guide the development of private and efficient ML models in healthcare, with the prospects of translating research efforts into real-world settings.

## 1 INTRODUCTION

Machine Learning (ML) and Deep Learning (DL) have shown great promise in many domains, leveraging the use of large datasets. Some notable contributions include *AlphaFold* (Jumper et al., 2021) for the prediction of protein structures and *Transformers* (Vaswani et al., 2017) for natural language processing. Healthcare is one of the domains in which ML is expected to provide substantial improvements in the delivery of patient care worldwide (WHO, 2021). Given the rapid growth in the number of models over the last couple of years (Ravì et al., 2016; Miotto et al., 2018; Kaul et al., 2022; Javaid et al., 2022), healthcare applications deserve special consideration considering the sensitive nature of the data that is required to train the models and the safety-critical nature of medical decision-making.

In this regard, real-world implementation of such models is still hampered by ethical and legal constraints. Legal frameworks have been developed and enforced to guarantee the transparency and privacy of ML-based healthcare solutions, such as the *Health Insurance Portability and Accountability Act (HIPAA)* in the United States (Gostin et al., 2009) and the *General Data Protection Regulation (GDPR)* in Europe (Voigt & Von dem Bussche, 2017). Therefore, there is a crucial need for Privacy-Preserving Machine Learning (PPML) in healthcare to enable the implementation of trustworthy systems in the future. The main goal of this review is to provide a comprehensive overview of state-of-the-art PPML in healthcare and encourage the development of new methodologies that tackle specific challenges relevant to the nature of the domain.

**Motivation.** There exist several related literature reviews that focus on a specific subset of PPML for healthcare. Several highlight recent advancements in federated learning (Xu et al., 2021; Ali et al., 2022; Joshi et al., 2022; Nguyen et al., 2022), cryptographic techniques (Zalonis et al., 2022), or security aspects of ML models, such as adversarial attacks (Liu et al., 2021). Existing review articles cover a wide range of applications related to health and input data modalities, ranging from IoT sensors to medical images (Qayyum et al., 2020). Compared to existing work, our review has three main contributions with the intent of bridging between research pertaining to ML for healthcare and cybersecurity. First, we distinguish between PPML for training and inference, i.e., *ML-as-a-service*.

---

*Equal contributions.

Second, we focus on state-of-the-art (SOTA) literature published in the last three years, considering the high proliferation of ML in healthcare and recent methodological advancements in ML and DL (e.g., network architectures, model pre-training, etc.). Third, we consider studies that develop or apply methodologies using two popular modalities based on publicly available datasets and state-of-the-art in ML for healthcare, namely medical images and data extracted from Electronic Health Records (EHR) (Kaul et al., 2022).Despite the use of other input modalities in medical applications, such as video (Ouyang et al., 2020) or text (Srivastava et al., 2019), our review exclusively focuses on medical images and EHR as they are the most prevalent input modalities in diagnostic and prognostic settings (Shehab et al., 2022). Lastly, although we acknowledge the importance of security for ML models, it is out of the scope of this paper since we primarily focus on privacy.

To this end, we review papers that meet the following inclusion criteria:

1. We include recently published work i.e., publication year $\geq$ 2020.
2. We include articles that focus on the application or development of PPML either for model training and/or inference, including but not restricted to homomorphic encryption, differential privacy, federated learning, and multi-party secure computation.
3. We include articles that consider clinical tasks involving medical images and/or EHR data.

In Section 2, we provide background knowledge about concepts and terminology concerning PPML. In Section 3, we provide an overview of the state-of-the-art pertaining to PPML for training (Section 3.1) and for inference (Section 3.2). Later in Section 4, we discuss open challenges and derive future directions. Finally, we provide concluding remarks in Section 5.

## 2 PRIVACY-PRESERVING MACHINE LEARNING: BACKGROUND & TERMINOLOGY

### 2.1 FEDERATED LEARNING

Since medical data is highly sensitive, data sharing is difficult, and subject to ethical restrictions and legal constraints if at all possible. Federated learning (FL) (McMahan et al., 2017) aims to overcome the challenges of data sharing by enabling collaborative training, which does not require that the involved parties share their training data. Therefore, the data remains private to each local node within the FL network, such that only the model updates are shared and integrated in a centralized model.

Federated averaging (McMahan et al., 2017) is the most common form of FL. In this setting, a centralized server is connected to $N$ entities, which have their own training data. The central server orchestrates the collaborative training process as follows: (1) the initial model is distributed amongst all entities, (2) each entity performs a training iteration on their local model using their own training data, typically one epoch, and shares its resulting model parameters with the central server, (3) the server averages the model parameters shared by all entities and distributes the resulting (averaged) model amongst all entities, and (4) steps (2) and (3) are repeated sequentially until a performance threshold or a specific number of training iterations is achieved. FL has proven to be very efficient in training models with strong performance, while avoiding the need for data sharing (McMahan et al., 2017). However, FL might be vulnerable to privacy issues such as reconstruction attacks (Liu et al., 2022), thus requiring that it is combined with other privacy-preserving methods to ensure robust privacy guarantees (Nguyen et al., 2022).

### 2.2 DIFFERENTIAL PRIVACY

Differential Privacy (DP) has its origins in statistical analysis of databases. Its main aim is to address the paradox of learning nothing about specific individuals, while learning useful information about the general population (Dwork et al., 2014). In the FL context, it is usually incorporated in the form of additive noise to model updates, either artificially or using a differentiable private optimizer, prior to transferring the updates from the entities to the central server (Abadi et al., 2016). The amount of artificial noise added is directly proportional to the degree of privacy desired (i.e., privacy budget) (Zhang et al., 2021c). DP can successfully make privacy attacks fail, such as reconstruction attacks,

as the added noise hinders the inference of actual knowledge about the training data by the attacker. However, adding too much noise (i.e., high privacy budget) can hamper learning and negatively impact the model accuracy (Chilukoti et al., 2022).

## 2.3 HOMOMORPHIC ENCRYPTION

In mathematics, the term *homomorphic* refers to the transformation of a given set into another while preserving the relation between the elements in both sets. Thus, Homomorphic Encryption (HE) refers to the conversion of plaintext into ciphertext while preserving the structure of the data. Consequently, specific operations applied to the ciphertext will provide the same results as if they were applied to the plaintext but without compromising the encryption (Acar et al., 2018). That is, the plaintext data is never accessed nor decrypted as the operations are directly applied to the encrypted data. The result of the transformations on the ciphertext can only be decrypted back to plaintext by the encryption key owner.

Despite the benefit of provable privacy guarantees, the range of operations available in HE is restricted to addition and multiplication i.e., *fully* homomorphic encryption. This limits the set and number of transformations applicable to the data and requires the use of approximations for more complex operations (e.g., HE-ReLU is the polynomial approximation of the ReLU function (Yue et al., 2021b)). This also significantly increases the computational time needed to process encrypted text compared to plaintext by several orders of magnitude (Popescu et al., 2021).

## 2.4 SECURE MULTI-PARTY COMPUTATION

Secure Multi-Party Computation (SMPC) (Goldreich, 1998) provides a framework in which two or more parties jointly compute a public function with their data while keeping the inputs private and hidden from other parties using cryptographic protocols. Most protocols used for SMPC with more than two parties are based on Secret Sharing (SS). In SS, a portion of the secret input is shared among a number of other parties. Most ML methods use Shamir's SS and additive SS (Singh & Shukla, 2021b). Although these methods are considered information-theoretic secure cryptosystems, recent studies show that leakage of global data properties can occur in some scenarios (Zhang et al., 2021a). While both FL and SMPC rely on collaborative training via knowledge sharing and keep the endpoint data private, their implementation differs significantly. SMPC involves cryptography and can be used for training and inference, whereas FL does not involve cryptography nor provides strong privacy guarantees, and is only used for model training.

## 3 OVERVIEW OF STATE-OF-THE-ART

Following the inclusion criteria described in Section 1, we summarize existing work on PPML for healthcare based on whether the work focuses on model training (Table 1) or model inference (Table 2). For each study (row) we describe several attributes. *Use case* provides a succinct summary of the objective of the study. *Model* reports the ML or DL architecture that was employed to model the task. *Medical datasets* summarizes the datasets that were used for model training and evaluation. Additionally, we use the * symbol to indicate the use of a private dataset. *ML task* describes the nature of the prediction task (e.g., binary or multi-class classification). *Input modality* reports the nature of the model's input data, which could either be *I* for medical images or *E* for EHR data. In the *Validation* column, we report whether the trained model was internally and/or externally evaluated, with ✔indicating the use of internal validation i.e., test set from the same distribution of the training data, and ✔✔indicating the assessment of the generalization of the model on an external test dataset. Lastly, *Metrics* lists the evaluation metrics used to describe the performance of the proposed model.

### 3.1 PRIVACY-PRESERVING TRAINING FOR HEALTHCARE

As observed in Table 1, the most commonly used privacy-preserving approach for model training is FL, either independently or in combination with DP. DP is added to increase the privacy of the FL training updates i.e., adding noise to the shared weights, thus making the system more robust to

Table 1: **Summary of PPML in healthcare for model training.** We summarize studies that focus on developing PPML in the context of model training. We group them based on the methodology considered, i.e. federated learning, homomorphic encryption, and differential privacy.

| Reference | Use case | Model | Medical dataset/s | ML task | Input modality | Validation | Metrics |
|---|---|---|---|---|---|---|---|
| **FEDERATED LEARNING** | | | | | | | |
| Dou et al. (2021) | COVID-19 Computed Tomography (CT) analysis | RetinaNet | Multi-institution lung CT data* | Object detection | I | ✔✔ | mAP, Specifity, Recall, AUROC |
| Field et al. (2022) | Cardiovascular admission after lung cancer treatment | Logistic regression | Multi-institution lung CT data* | Risk prediction | I+E | ✔ | AUROC, C-index |
| Lee & Shin (2020) | FL benchmarking and reliability in healthcare | Neural Network, LSTM, CNN | MIMIC-III, PhysioNet ECG | Mortality prediction, Multi-class classification | I+E | ✔ | AUROC, AUPRC, F1-score |
| Yang et al. (2022) | FL benchmarking and monetary cost in healthcare | Transformer, EfficientNet-B0, ResNet-NC-SE | eICU, ISIC19, HAM10000, PhysioNet ECG | Mortality prediction, Length of stay, Discharge time, Acuity prediction | I+E | ✔ | AUROC, AUPRC |
| Sadilek et al. (2021) | FL benchmarking vs. centralized learning in healthcare | Logistic regression, Neural Network, Generalized linear model | UCI Heart failure, MIMIC-III, Malignancy in SARS-CoV-2 infection | Risk prediction | E | ✔ | AUROC |
| Loftus et al. (2022) | COVID-19 detection | DenseNet | Multi-institution COVID-19 X-ray* | Binary classification | I | ✔✔ | AUROC, AUPRC |
| Wolff et al. (2022) | Coronary artery calcification (CAC) forecast | Random Forest | CAC risk factors* | Risk prediction | E | ✔ | Recall, Specificity |
| Wang & Zhou (2022) | Cancer inference via gene expression | Gradient Boosting Decision Tree | iDASH 2020 | Multi-class classification | E | ✔ | Accuracy, AUC, Recall, Precision, F1-score |
| Islam et al. (2022a) | Diabetic kidney risk prediction | Logistic regression, MLP | CERNER Health Facts | Risk prediction | E | ✔ | F1-score |
| Deist et al. (2020) | Lung cancer post-treatment 2-year survival | Logistic regression | Multi-institution lung cancer EHR* | Mortality prediction | E | ✔ | RMSE, Accuracy, AUROC |
| Park et al. (2021) | COVID-19 detection | Transformer with DenseNet, TransUNet and RetinaNet | Multi-institution COVID-19 X-ray (public and private datasets) | Multi-task: classification, segmentation, object detection | I | ✔✔ | AUC, mAP, Dice coefficient |
| Yan et al. (2023) | Multiple medical prediction tasks | Self-supervised vision transformer | COVID-19 X-ray, Kaggle Diabetic Retinopathy, Dermatology ISIC | Binary/multi-class classification, Object detection | I | ✔✔ | Accuracy, F1-score |
| **HOMOMORPHIC ENCRYPTION** | | | | | | | |
| Boulila et al. (2022) | COVID-19 detection | MobileNet-V2 | COVID-19 X-ray | Multi-class classification | I | ✔ | Accuracy, Recall, Precision, F1-score |
| Ma et al. (2020) | Heart and thyroid disease classification | XGBoost | UCI Heart Disease, Kaggle Hypothyroid | Binary classification | E | ✔ | Accuracy |
| Paul et al. (2021) | Intensive Care Unit patient outcome | LSTM | MIMIC-III | Binary classification | E | ✔ | Recall, AUROC, Precision |
| Chen et al. (2022) | Dermatology diagnostics | SVM | UCI Dermatology | Multi-class classification | E | ✔ | Accuracy |
| Baruch et al. (2022) | COVID-19 detection | AlexNet, SqueezeNet | COVID-19 X-ray, COVID-19 CT | Multi-class classification | I | ✔ | Accuracy, F1-score |
| **DIFFERENTIAL PRIVACY** | | | | | | | |
| Zhang et al. (2021b) | Thoracic pathology detection | DenseNet-121 | CheXpert | Multi-class classification | I+E | ✔ | AUROC, Accuracy |
| Chilukoti et al. (2022) | COVID-19 detection | EfficientNet-B2 | COVID-19 X-ray | Binary classification | I | ✔ | Accuracy |
| Suriyakumar et al. (2021) | Multiple medical prediction tasks | CNN, DenseNet-121, Logistic regression, GRU-D | MNIST NIH Chest X-ray, MIMIC-III | Binary, Multi-class classification | I+E | ✔ | AUROC |

privacy threats, such as reconstruction attacks by an external actor intercepting the communication channel or an *honest-but-curious* central server (Nguyen et al., 2022).

The second most commonly investigated approach for private training is HE, which leverages encryption schemes to provide privacy with provable mathematical guarantees. However, as described in the previous section, training ML models on encrypted data significantly increases the computational complexity and the processing overhead by several orders of magnitude (Wibawa et al., 2022; Zhang et al., 2022). It also adds noise to the training process due to the approximations of activation functions, especially in large models.

The third most common approach is standalone DP, which is less computationally demanding and provides strong privacy guarantees. However, the increase in privacy guarantees is negatively correlated with model accuracy, as it is associated with an increase in the quantity of noise applied. Therefore, the trade-off between privacy (i.e., privacy budget) and model accuracy is a relevant factor to take into account for the inclusion of DP in any ML solution. There are other PPML ap-

Table 1 Continued: **Continued summary of PPML in healthcare for model training.** We summarize here studies that use a combination of federated learning and other privacy-preserving techniques, blockchain, Secure Multi-Party Computation (SMPC), image encryption, and image modification.

| Reference | Use case | Model | Medical dataset/s | ML task | Input modality | Validation | Metrics |
|---|---|---|---|---|---|---|---|
| **FEDERATED LEARNING + DIFFERENTIAL PRIVACY** | | | | | | | |
| Islam et al. (2022b) | Cardiomyopathy risk prediction | Random Forest, Naive Bayes | iDASH 2021, Breast Cancer TCGA | Risk prediction | E | ✔ | AUROC |
| Kerkouche et al. (2021) | In-hospital mortality prediction | CNN | Premier Healthcare Database* | Mortality prediction | E | ✔ | AUROC, Overhead |
| Dayan et al. (2021) | COVID-19 patient triage | ResNet-34 DeepCrossNet | Multi-institution chest x-ray and EHR* | Risk prediction | I+E | ✔✔ | AUROC, Recall, Specificity |
| **BLOCKCHAIN** | | | | | | | |
| Zerka et al. (2020) | Distributed training | ResNet-18 | NSCLC-Radiomics | Binary classification | I | ✔✔ | AUROC |
| Warnat-Herresthal et al. (2020) | Disease classification | Neural Network | Blood transcriptomes* | Binary classification | E | ✔ | Accuracy |
| **FEDERATED LEARNING+HOMOMORPHIC ENCRYPTION** | | | | | | | |
| Wibawa et al. (2022) | COVID-19 detection | CNN | COVID-19 X-ray | Binary classification | I | ✔ | Accuracy, Recall Precision, F1 score, Execution time |
| **FEDERATED LEARNING+HOMOMORPHIC ENCRYPTION+SMPC** | | | | | | | |
| Zhang et al. (2022) | Skin cancer classification | CNN | HAM10000 | Multi-class classification | I | ✔ | Accuracy, Overhead |
| **SMPC** | | | | | | | |
| Hong et al. (2020) | Tumor detection | Logistic regression | iDASH 2019 | Binary classification | E | ✔ | Accuracy, Overhead |
| **IMAGE ENCRYPTION** | | | | | | | |
| Huang et al. (2022) | Brain tumor, COVID-19 | DenseNet-121, XceptionNet | MRI Brain Tumor, COVID-19 X-ray | Multi-class classification | I | ✔ | F1-score |
| **IMAGE MODIFICATION** | | | | | | | |
| Montenegro et al. (2021) | Glaucoma recognition | VGAN-based CNN | Warsaw-BioBase Disease-Iris v2.1 | Binary classification | I | ✔ | F1-score, Accuracy |

proaches for model training that have been evaluated in related work, including the addition of a blockchain ledger to avoid the centralization of training (i.e., fully distributed learning), image modification to increase data privacy in the context of model explainability, and SMPC as an alternative encryption scheme to HE.

Most of the reviewed studies use a single source of input data i.e., image or EHR and only one medical dataset. Although some studies train their models on several datasets, including popular computer vision benchmarks, the vast majority restrict their evaluation to one input modality from the same dataset.

This limits the generalization of the results and neglects the potential improvement in predictive performance that could result from combining different data sources in multi-modal learning settings (Ramachandram & Taylor, 2017). Furthermore, most studies perform internal validation, such that the test sets are from the same distribution as the training dataset. This is generally a common challenge in healthcare applications considering distribution shifts across different hospitals, for example due to differences in patient demographics. Finally, most existing work focuses on convolutional neural networks to handle computer vision tasks. However, validation schemes and metrics reported are not consistent, making the comparison among them very difficult. Due to these reasons and the lack of medical benchmark datasets, a fair comparison of the approaches is difficult, and therefore we do not assess performance metrics results in this review and defer it to future work.

## 3.2 PRIVACY-PRESERVING INFERENCE FOR HEALTHCARE

We now focus on the literature employing PPML methods for inference, as summarized in Table 2. We frame PPML for inference as providing private *machine-learning-as-a-service* (MLaaS) or *inference-as-a-service* (IaaS) (Lins et al., 2021). In this scenario, a model with strong performance is controlled by a single party (i.e., model owner), and other external parties (i.e., clients) would like the model to perform inference on their own data. The external parties can share data samples with the model owner and their predictions are sent back. Due to legal and/or ethical con-

Table 2: **Summary of PPML in healthcare for model inference.** We summarize studies that focus on developing PPML in the context of model inference. We group them based on the methodology considered, i.e. homomorphic encryption, combination of federated learning and Secure Multi-Party Computation (SMPC), differential privacy and SMPC, federated learning with blockchain and SMPC, and finally federated learning with differential privacy and homomorphic encryption.

| Reference | Use case | Model | Medical dataset/s | ML task | Input modality | Validation | Metrics |
|---|---|---|---|---|---|---|---|
| **HOMOMORPHIC ENCRYPTION** | | | | | | | |
| Yue et al. (2021a) | Breast and cervical cancer classification | Convolutional LSTM | Cervigram Image, BreaKHis | Binary, Multi-class classification | I | ✔ | AUROC |
| T'Jonck et al. (2022) | Breast cancer classification | Neural Network, SVM | UCI IRIS, UCI Breast Cancer | Binary, Multi-class classification | E | ✔ | Accuracy, Privacy budget, Overhead |
| Sarkar et al. (2022) | Cancer inference via gene expression | SVM, Logistic regression, Neural Network | iDASH 2020 | Multi-class classification | E | ✔✔ | Accuracy, AUROC |
| Vizitiu et al. (2020) | Coronary angiography view classification | CNN | X-ray coronary angiography* | Binary, Multi-class classification | I | ✔ | Accuracy |
| **FEDERATED LEARNING + SMPC** | | | | | | | |
| Ziller et al. (2020), Kaissis et al. (2021)+ | Paediatric chest X-ray classification | ResNet-18 | Chest X-ray* | Multi-class classification | I | ✔✔ | AUROC, Latency |
| **DIFFERENTIAL PRIVACY + SMPC** | | | | | | | |
| Singh & Shukla (2021a) | Pneumonia detection | CNN, VGG-16 | Kaggle X-ray Pneumonia | Binary classification | I | ✔ | Accuracy |
| Jarin & Eshete (2021) | Accuracy-privacy trade-off analysis | Neural Network | Kaggle IDC, MIMIC-III | Binary, Multi-class classification | I | ✔ | Accuracy, Recall Precision, Privacy |
| **FEDERATED LEARNING + BLOCKCHAIN + SMPC** | | | | | | | |
| Kasyap & Tripathy (2021) | Multiple medical image datasets classification | CNN | MedMNIST (CXR, Breast, Hand, ChestCT, Abdomen, HeadCT) | Multi-class classification | I | ✔ | Accuracy |
| **FEDERATED LEARNING + DIFFERENTIAL PRIVACY + HOMOMORPHIC ENCRYPTION** | | | | | | | |
| Gopalakrishnan et al. (2021) | Multiple medical image datasets classification | CNN | MedMNIST (Pneumonia Breast, Retina, Blood) | Multi-class classification | I | ✔ | Accuracy, Execution time, Bandwidth |

+ Kaissis et al. (2021) is an extension of Ziller et al. (2020).

straints related to privacy, clients cannot disclose their data with the model owner, thus requiring the use of PPML to maintain the privacy of the data they wish to share.

Compared to the number of studies addressing PPML for training, a relatively fewer number have explored PPML for inference. Most studies within the theme of PPML for inference, focus on the deployment of the trained model as a service and its use by third parties. The most common approach for delivering PPML IaaS is HE, which ensures with provable mathematical guarantees that neither the model owner nor any intermediate party are able to inspect the original data nor the detection result i.e., both are encrypted and can only be decrypted by the data owner. Another common approach is SMPC, which also leverages encryption schemes, being used in combination with other privacy-preserving collaborative approaches such as FL, DP and blockchain.

Similar to PPML for training, most studies here use a single source of input data (i.e., images in most cases), neglecting many other diverse medical modalities of varying characteristics. The lack of use of benchmark medical datasets and inconsistent validation schemes and metrics hinders the generalization of the proposed approaches.

## 3.3 OPEN CHALLENGES

**There is no *one-size-fits-all* PPML approach for model training or inference by design.** We observe that previous work pick and choose PPML approaches based on the intended clinical use case. Currently, there is no consensus on what different "privacy models" look like in healthcare. Since the methodology depends on the use case, we also observe a clear trade-off between privacy and accuracy, based on the availability of computational resources. For instance, standalone FL is computationally faster than HE, but it does not provide strong privacy guarantees. On the other hand, HE and DP can provide strong privacy guarantees but they add noise to the model both for private training and private inference resulting in less accurate solutions. For HE, this is especially critical

for model training where successive layers of approximations are needed to perform operations that are not supported, such as *softmax*, or that are computationally inefficient, such as max pooling. In general, encryption-based options are provably secure but computationally inefficient, since they increase the processing overhead of training using cyphertext compared to plaintext data.

Additionally, the availability of computational resources is a decisive factor in choosing a particular PPML methodology. For instance, in the HE scenario, sending data over a communication channel does not require infrastructure for model training but still requires handling the encryption/decryption process appropriately. In the FL context, it requires that the entity has allocated resources for model training.

**The centralization of model training in FL poses an additional security threat.** Relying on a single central server entails a single point of failure that is highly susceptible to security attacks such as Denial-of-Service. Although blockchain has been proposed to achieve fully distributed training and mitigate this threat, it increases the complexity of the information technology infrastructure significantly, requiring dedicated resources for the implementation of the distributed ledger and modeling framework.

**Most existing work use a single dataset and do not conduct external validation, thus arising concerns about the generalization of the results.** We observe that existing work focus on a limited set of medical datasets. Additionally, some work only evaluate their solutions on computer vision benchmark datasets (e.g., *MNIST* or *CIFAR-10*) inferring that good performance on these datasets will provide similar results on medical image data (Festag & Spreckelsen, 2020; Onesimu & Karthikeyan, 2020). However, this assumption is not empirically supported by work that uses both medical and non-medical datasets (Suriyakumar et al., 2021; Zhang et al., 2021b; Gopalakrishnan et al., 2021; Jarin & Eshete, 2021; Vizitiu et al., 2020) and must, therefore, be avoided.

**MLaaS for healthcare has not been explored thoroughly.** As demonstrated by the limited literature on this topic, we observe that the literature is highly skewed towards PPML for training. Considering disparities in technical capabilities and expertise, information technology resources, and availability of data across medical institutions, the case in which an entity does not have enough resources to perform model training independently is highly likely. Thus, the usage of third-party models as inference systems that can run on proprietary data is a prominent scenario that has not been thoroughly explored and should be considered in future research. MLaaS can provide access to models with strong performance, enabling full preservation of data privacy using PPML methods. This makes it a more efficient solution for small-scale or low-resource medical entities to access and leverage third-party knowledge.

## 4 FUTURE RESEARCH DIRECTIONS

### 4.1 COMPREHENSIVE EVALUATION ON DIVERSE MEDICAL DATASETS

For the sake of comparison and generalization of results, studies should complement their internal dataset evaluation with additional extensive evaluation on benchmark medical datasets. This is due to the fact that most of the existing work use a single dataset and do not perform external validation. The number of studies that use external datasets for validation is marginal. Only 9 out of the 40 studies considered validated their results with an independent test set. This hampers model generalization and hinders performance comparison among approaches built for the same medical task. For benchmarking, we suggest *MedMNIST* (Yang et al., 2023), which contains curated datasets for different medical tasks and modalities. Therefore, similar to *MNIST* or *CIFAR-10* for computer vision models, this medical dataset could be employed as a common benchmark for medical applications.

### 4.2 MULTI-MODAL MODELS

Current advances in ML for healthcare are moving towards multi-modal learning, where several sources of information are combined to improve performance (Ramachandram & Taylor, 2017; Soenksen et al., 2022). This approach not only tends to provide better performance but also ensures a comprehensive understanding of the different physiological variables involved in studying and modeling the development of human biology and pathology. As observed in Sections 3.1 and 3.2, most work is restricted to a single modality. To develop robust and strong ML models, the use of

different data sources to develop multi-modal systems is paramount. Notwithstanding that, the use of more clinical data entails more privacy concerns (e.g., individuals may be identified using correlated data) and requires more training resources due to increased model complexity. Therefore, additional privacy and computational constraints must be considered in the design of these algorithms.

### 4.3 MACHINE LEARNING AS A SERVICE (MLaaS)

The deployment of PPML within MLaaS is a very promising opportunity to access strong proprietary models by less resourceful institutions. Indeed, one of the main objectives of ML in healthcare is to develop efficient and scalable solutions that improve healthcare delivery. In addition to lack of resources, the deployment of these systems in medical settings can also be highly challenging (Kreuzberger et al., 2022; Wiesenfeld et al., 2022). The development of MLaaS is significantly less investigated than PPML for model training. Therefore, further research on this topic is required to provide secure, private and efficient data sharing between third-party model providers and client institutions. Reducing obstacles for clinical institutions to access powerful inference systems could lead to a major improvement in healthcare delivery across regions, bypassing physical barriers. It can also lead to an increase in the confidence and widespread adoption of ML in healthcare. It is important to note that the success of MLaaS is dependent on improvements in model generalizability and fairness in external datasets.

### 4.4 INTEGRATION OF SOTA AND ADVANCES IN DEEP LEARNING

Future work should also investigate the integration of recent advances in DL and ML models in healthcare, considering that most of the current PPML work focuses on convolutional neural networks. For instance, the *Transformer* architecture and its variants (Dosovitskiy et al., 2020), which are considered the current SOTA for many computer vision or natural language processing tasks, are only adopted by Park et al. (2021), Yang et al. (2022), and Yan et al. (2023) in the current related literature. Adopting SOTA architectures can take advantage of the latest advances in research, both in terms of optimizing hardware and software, to maintain performance improvements in clinical prediction tasks.

### 4.5 GLOBAL AND LOCAL EXPLAINABILITY

Transparency and model explainability are essential for trustworthy artificial intelligence (OECD, 2023b). However, PPML methods, such as data encryption or noise addition, hinder global model and local prediction explainability. The collision between two key principles for trustworthy artificial intelligence, secure and PPML (OECD, 2023a) and explainability, highlights an important research problem that is currently under-investigated. Only Montenegro et al. (2021) attempt to address this problem, which should encourage future work in this research direction.

## 5 CONCLUSION

In this paper, we introduce and summarize recent literature concerning PPML for model training and inference in the healthcare domain. We highlight trends, challenges and promising future research directions. In conclusion, we recognize the lack of consensus when it comes to defining the requirements of privacy-preserving frameworks in healthcare. This requires collaboration between machine learning scientists, healthcare practitioners, and privacy and security experts. From the perspective of advancing ML approaches, we encourage researchers to perform comprehensive evaluation of proposed algorithms on diverse medical datasets to increase generalization, to investigate the constraints of PPML in multi-modal learning settings, to further consider the promise of MLaaS in healthcare as a catalyst for improved healthcare delivery, and to adopt state-of-the-art advances in deep learning architectures to enhance model performance. Our suggestions aim to address research gaps and guide future research in PPML to facilitate the future adoption of trustworthy and private ML for healthcare.

**Acknowledgements.** This work was supported by the NYUAD Center for Interacting Urban Networks (CITIES), funded by Tamkeen under the NYUAD Research Institute Award CG001, and the Center for Cyber Security (CCS), funded by Tamkeen under NYUAD RRC Grant No. G1104.

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
