# OpenReview forum: "Privacy-preserving machine learning for healthcare: open challenges and future perspectives"
_ICLR.cc/2023/Workshop/TML4H — ICLR 2023 Workshop TML4H Oral_

### Official Review · Reviewer_tumt · 2023-03-03
**The paper provides informative summaries and valuable insights of  privacy-preserving machine learning for healthcare.**

**Rating:** 8
**Confidence:** 3

**Review:**

The paper presents a review of privacy-preserving machine learning for healthcare. The topic covers a range of PPML scenarios and privacy-enhancing frameworks, and the authors make a comprehensive investigation on the related state-of-the-art publications since 2020. The authors also give their views on the open challenges and future directions. The paper starts with a high-level summary of the importance of PPML and expands with comprehensive aspects, and their insights are valuable. The paper reads smoothly.

Comments:
1. the author can slightly expand Section 2.4 to differentiate multi-party secure computation and federal learning
2. the author can summarize the benefits of PPML methods (improvements/advantages) in each listed SOTA publications, which helps readers understand how PPML methods fit specific clinical cases.

---

### Official Review · Reviewer_8Wti · 2023-03-04
**Review of Paper25**

**Rating:** 7
**Confidence:** 4

**Review:**

**Summary of The Paper**:

This paper provides a comprehensive review of Privacy-Preserving Machine Learning (PPML) technologies for healthcare. It introduces the background and widely used technologies in PPML and then discusses the state-of-the-art methods for each category. The authors also discuss existing challenges and future directions.

**Strengths**:

1.This paper is well-written and easy to follow.

2. The authors provide a thorough overview of the current used technologies and also mention the pros and cons for each one.

3. The authors' suggestions for future research directions are aligned with previously discussed challenges.


**Weaknesses**:

Here are some suggestions that could contribute to a more comprehensive review:
1. In sections 3.1 and 3.2, it would be beneficial if the authors provided some quantitative analysis of the results for each method.

2. In section 3.1, since the authors mentioned that computational resources in Homomorphic Encryption are a problem, it may be helpful to include more related information.

3. The paper mainly focuses on image and HER data. It could be improved by including more related literature on other modalities, such as videos.

4. In section 4.4, the authors state that "Transformer architecture and its variants are only adopted by Yang et al. (2022) in the current related literature". However, it is likely that there are other state-of-the-art methods based on Transformer architecture that could be included in the review to provide a more comprehensive overview of the field. For example, "Federated Split Vision Transformer for COVID-19 CXR Diagnosis using Task-Agnostic Training", "Benchmarking Differential Privacy and Federated Learning for BERT Model", and "Label-Efficient Self-Supervised Federated Learning for Tackling Data Heterogeneity in Medical Imaging" all employ Transformer-based Federated Learning methods.

---

### Meta-Review · Area_Chair_ywuA · 2023-03-05

**Recommendation:** Accept (Oral)
**Confidence:** 5

**Metareview:**

This paper provides a thorough overview of the current Privacy-Preserving Machine Learning (PPML) technologies. Overall, all reviewers enjoy reading this paper, and find that it is well-written and easy to follow. Some minor concerns have been raised regarding the absence of quantitative analysis for certain methods and the narrow focus on image and HER data, without including other modalities such as videos.

The authors are highly encouraged to address the above (minor) concerns in the final version.